# Immunotherapy Predictive Molecular Markers in Advanced Gastroesophageal Cancer: MSI and Beyond

**DOI:** 10.3390/cancers13071715

**Published:** 2021-04-05

**Authors:** Robin Park, Laercio Lopes Da Silva, Anwaar Saeed

**Affiliations:** 1Department of Medicine, MetroWest Medical Center/Tufts University School of Medicine, Framingham, MA 01702, USA; robin.park@mwmc.com (R.P.); laercio.lopesdasilva@mwmc.com (L.L.D.S.); 2Department of Medicine, Division of Medical Oncology, Kansas University Medical Center, Kansas City, KS 66205, USA

**Keywords:** immunotherapy, gastroesophageal cancer, biomarkers, programmed death-1/ligand-1, microsatellite instability/mismatch repair, tumor-mutational burden

## Abstract

**Simple Summary:**

Biomarker research for immunotherapy in gastroesophageal cancer has been rapidly developing in parallel with the growing use of immune checkpoint inhibitors. Although several biomarkers have been approved for use in the clinical setting they are imperfect in predicting responses. Several novel biomarkers are currently being studied and demonstrate potential for application in the clinical setting. Future research should determine the ability of immunotherapy biomarkers to predict response and synergy in combination therapy.

**Abstract:**

Advanced gastroesophageal cancer (GEC) has a poor prognosis and limited treatment options. Immunotherapy including the anti-programmed death-1 (PD-1) antibodies pembrolizumab and nivolumab have been approved for use in various treatment settings in GEC. Additionally, frontline chemoimmunotherapy regimens have recently demonstrated promising efficacy in large phase III trials and have the potential to be added to the therapeutic armamentarium in the near future. There are currently several immunotherapy biomarkers that are validated for use in the clinical setting for GEC including programmed death ligand-1 (PD-L1) expression as well as the tumor agnostic biomarkers such as mismatch repair or microsatellite instability (MMR/MSI) and tumor mutational burden (TMB). However, apart from MMR/MSI, these biomarkers are imperfect because none are highly sensitive nor specific. Therefore, there is an unmet need for immunotherapy biomarker development. To this end, several biomarkers are currently being evaluated in ongoing trials with some showing promising predictive potential. Here, we summarize the landscape of immunotherapy predictive biomarkers that are currently being evaluated in GEC.

## 1. Introduction

GEC comprises a group of solid tumors arising from the esophagus, gastroesophageal junction, and stomach, including gastric cancer (GC), esophageal and gastroesophageal junction adenocarcinoma (EAC), and esophageal squamous cell carcinoma (ESC). GEC accounts for an estimated 45,820 new cases per year and 26,710 deaths per year in the United States [1]. The treatment of GEC consists of (i) surgical resection or definitive chemoradiation in localized tumors; (ii) surgical resection with perioperative chemoradiation or chemotherapy in locally advanced tumors; and (iii) palliative systemic therapy in advanced or metastatic tumors. Despite developments in diagnosis and treatment, the prognosis of GEC in advanced or metastatic stages remains very poor, with a five-year relative survival rate of 5–6% in the United States [2,3].

The standard palliative treatment in the first line setting for GEC is a combination cytotoxic chemotherapy with fluoropyrimidine and cisplatin with or without trastuzumab based on HER-2 status [4,5]. Recent developments in immunotherapy in GEC have led to the approval of pembrolizumab in the second line setting in PD-L1 + ESC and third line setting in PD-L1 + GEC and nivolumab in the second line setting in ESC regardless of PD-L1 expression. Additionally, a front-line chemoimmunotherapy regimen consisting of pembrolizumab with fluropyrimidine and cisplatin and nivolumab with FOLFOX/XELOX were given Food and Drug Association (FDA) priority review designation status for EC on 17 December 2020 based on the KEYNOTE-590 and CheckMate-649 studies [6,7]. However, response rates for immunotherapy as single agent and combination could be further optimized when tailored to the right subgroup of patients. Therefore, improvement in patient selection and biomarker use is a great unmet need.

Whereas ESC and EAC differ in the underlying etiology and the characteristic genomic and molecular alterations, EAC and GC have significant overlaps and are grouped into a single disease spectrum based on several large scale, sequencing and bioinformatic analyses. The Cancer Genome Atlas (TCGA) analysis conducted in the U.S. demonstrated that EAC and GC can be classified into four distinct molecular subtypes: (i) the MMR deficient/MSI-high (dMMR/MSI-H) subgroup; (ii) Epstein-Barr virus (EBV) subgroup, which is characterized by tumors with high PD-L1 or PD-L1 expression; (iii) chromosomal instability subgroup; and (iv) genomically stable subgroup [8,9]. Also, the Asian Cancer Research Group (ACRG) conducted a similar analysis with an analogous classification [10]. Additionally, the Oesophageal Cancer Clinical and Molecular Stratification (OCCAMS) analysis in Europe has established a distinct classification with three subtypes: (i) tumors with a defective homologous recombination (HR) pathway, (ii) tumors with common T > G mutations characterized by high TMB (TMB-H), and (iii) tumors with a dominant C > A/T mutational pattern [11]. Such classification schemes represent useful tools that can guide rational treatment selection based on tumor immunobiology.

Immunotherapy biomarkers currently approved for clinical use in GEC are the tumor-agnostic markers, MMR/MSI status and TMB, and PD-L1 expression, measured using the combined positive score (CPS). Additionally, there are several putative biomarkers currently under investigation in GEC including EBV, DNA damage response (DDR) gene alterations as well as gene expression profiles (GEP) that indicate active inflammation in the tumor microenvironment (TME) Figure 1. Herein we briefly describe and summarize the established and putative IO biomarkers of relevance in GEC and discuss the future direction of research in this area.

## 2. Immunotherapy Trials in Gastroesophageal Cancer

Inhibitory immune checkpoint axes such as PD-1/PD-L1 or B7/CTLA-4 thwart anti-tumor immune responses by suppressing cell mediated cytotoxicity against tumor cells [12]. Thus, checkpoint inhibition using anti-PD-1 or -PD-L1 or -CTLA-4 monoclonal antibodies elicits antitumor immune responses, leading to potentially durable treatment responses. 

In the KEYNOTE-012 phase Ib trial, pembrolizumab demonstrated promising efficacy with an objective response rate (ORR) of 22% and a median duration of response (mDOR) of 40 months in patients with PD-L1-positive tumors, while the KEYNOTE-28 phase I/II trial showed an ORR of 30% and a mDOR of 15 months (Table 1) [13,14]. More importantly, the open-label, multicohort, KEYNOTE-059 phase II trial enrolled 259 patients with progression on two prior lines of systemic therapy to evaluate pembrolizumab efficacy and safety. Among the enrolled patients, 143 (55%) were PD-L1-positive with a CPS of 1 or higher and MMR proficient/microsatellite stable or MSI status undetermined. In these patients, pembrolizumab demonstrated an ORR of 13.3% and durable responses longer than 6 months in 58% of patients and 12 months in 26% of patients. The toxicity profile was comparable to that previously reported in patients with non-small cell lung cancer or melanoma treated with pembrolizumab [15]. Based on this trial, pembrolizumab was granted FDA approval in 2017 in GEC for patients with tumors with a PD-L1 CPS of 1 or higher in the third line setting [16]. Additionally, these results eventually led to the approval of pembrolizumab along with the PD-L1 CPS score as the companion diagnostic. 

Pembrolizumab was subsequently evaluated in the second line setting in the KEYNOTE-061 trial. In this open-label, phase III randomized trial enrolled patients who progressed on first line chemotherapy regardless of PD-L1 status to compare pembrolizumab to standard of care paclitaxel. Among the 592 patients enrolled into the study, 67% (326) of patients had CPS of 1 or higher. The study did not meet its primary endpoints of improved median OS and PFS in patients with CPS of 1 or higher. However, a post hoc, subgroup analysis showed that in patients with CPS of 10 or higher (*N* = 108), the median OS was prolonged in pembrolizumab compared to paclitaxel, suggesting potential benefit in this subpopulation [17].

In the frontline setting, pembrolizumab was evaluated in the KEYNOTE-062 trial. This phase III, multicenter, randomized trial assigned 763 treatment-naïve patients to either pembrolizumab with chemotherapy (cisplatin with 5-flurouracil or capecitabine), pembrolizumab alone, or standard of care chemotherapy with placebo. The data showed pembrolizumab alone or pembrolizumab with chemotherapy were not superior to chemotherapy in patients with CPS of 1 or higher. Additionally, pembrolizumab with chemotherapy was not superior to chemotherapy in patients with CPS of 10 or higher. While pembrolizumab alone demonstrated improved mOS compared to chemotherapy in patients with CPS of 10 or higher (HR 0.69, 95% CI 0.49–0.97), this analysis was not deemed significant as it was not statistically tested. The ORR and progression-free survival (PFS) of pembrolizumab alone was significantly worse compared to chemotherapy in both CPS of 1 or higher and 10 or higher (ORR 15% vs. 25%; PFS 2.0 vs. 2.9 months) [18].

Nivolumab is an anti-PD-1 antibody that was approved for use on 22 September 2017 for use in the second line setting in Japan regardless of PD-L1 expression [19]. The anti-tumor activity of nivolumab was first demonstrated in GEC in the CheckMate-032 phase I/II trial which assigned 160 patients treated with 1 or more prior lines of systemic therapy to nivolumab (3 mg/kg), nivolumab (1 mg/kg) with ipilimumab (3 mg/kg), or nivolumab (3 mg/kg) with ipilimumab (1 mg/kg). Data showed that nivolumab had an ORR of 12% and a 12-month overall survival (OS) of 39% and nivolumab with ipilimumab had an ORR of 24% and a 12-month OS 35% [20]. Subsequently, the ATTRACTION-2, phase III, randomized controlled trial was conducted in several Asian countries. In this trial, the investigators assigned 493 patients with GEC who progressed on two or more lines of prior therapy to nivolumab or placebo. Results showed that nivolumab prolonged OS over placebo (1-year OS rate 27% vs. 11%). Additionally, long term follow-up data show striking two-year survival rates of 10.6% in nivolumab compared to 3.2% in placebo [21]. The results of ATTRACTION-2 trial supported the approval of nivolumab in Japan for use in the second line setting.

Several studies have also tested the safety and efficacy of anti-PD-L1 therapies with or without anti-CTLA-4 therapy in GEC. In the JAVELIN Gastric 300, phase III, randomized, controlled trial, avelumab (an anti-PD-L1 antibody) was compared to chemotherapy in the third line setting. In this study, avelumab did not meet its prespecified primary endpoints including OS and PFS. Notably, avelumab showed a better toxicity profile compared to chemotherapy, which suggested that it may have a role as maintenance therapy following frontline chemotherapy [22]. However, the subsequent JAVELIN Gastric 100 trial demonstrated no significant mOS difference between avelumab maintenance and continued chemotherapy (10.4 vs. 10.9 months) [23]. Additionally, a phase I/II trial demonstrated that durvalumab (an anti-PD-L1 antibody) with tremelimumab (an anti-CTLA-4 antibody) was associated with some activity (ORR 7.4%, 12-month OS 37%), suggesting the need to further evaluate this strategy in pretreated GEC patients [24].

Single-agent anti-PD-1 therapies have also been evaluated in trials consisting only of EC patients. In the phase II KEYNOTE-180 trial, 121 patients with EAC or ESC who progressed on two or more prior lines of therapy received pembrolizumab. Data showed an ORR of 9.9% among all patients and 13.8% among patients with CPS 1 or higher [25]. Subsequently, pembrolizumab was compared to chemotherapy in the second-line setting in the randomized, controlled, phase III KEYNOTE-181 trial that enrolled 628 patients with ESC and EAC. Among all patients, mOS was not prolonged (mOS 7.1 vs. 7.1 months). A subgroup analysis of the study, however, did demonstrate prolonged mOS in ESC patients with CPS 10 or higher (10.3 vs. 6.7 months; HR 0.64, 95%CI 0.46–0.90) [26]. Based on these results, pembrolizumab was approved for use on July 30, 2019 for use in the second line setting for ESC patients with CPS of 10 or higher [27].

Additionally, the long-term follow-up results of the ATTRACTION-1 trial were recently published for the open-label, single-arm, multicenter phase II trial conducted in Japan in ESC patients who progressed on first line chemotherapy. The results reported for patients who had a minimum of five years of follow up showed an ORR of 17%, mOS of 10.8 months and mPFS of 1.5 months [28]. Based on the results of this trial, the ongoing ATTRACTION-3 trial is comparing nivolumab to chemotherapy in chemo-refractory ESC. The three-year follow-up results of this open-label, randomized phase III trial show an mOS benefit of nivolumab compared to chemotherapy (10.9 vs. 8.5 months; HR 0.79, 0.64–0.97). The 36-month OS rates were nearly doubled in the nivolumab group (15.3% versus 8.7%) [29]. Based on these results, nivolumab was approved by the FDA on 10 June 2020 in ESC patients who progressed on first line chemotherapy regardless of PD-L1 expression [30].

Upfront chemotherapy with checkpoint inhibition recently demonstrated positive results in GEC in several clinical trials. The KEYlargo, phase II, single-arm trial evaluating first line pembrolizumab with CapeOx in GEC recently reported interim results showing an excellent ORR of 72.7%, mPFS of 7.6 months, and mOS of 15.8 months with a grade 3–4 treatment-related adverse events of 44% [31]. Furthermore, the ongoing ATTRACTION-4, phase II/III trial is evaluating nivolumab with chemotherapy (oxaliplatin with S-1 or capecitabine) in Asian patients. The initial part of this study (phase II) demonstrated a promising ORR of 65% and mPFS of 9.5 months [32]. Importantly, the interim results of two landmark phase III trials evaluating frontline anti-PD-1 therapy with chemotherapy were reported at ESMO 2020. In the randomized, controlled phase III CheckMate-649 trial, 1581 patients were assigned to nivolumab with chemotherapy or chemotherapy alone. The primary endpoint of this trial was a dual primary endpoint of OS and PFS in patients with CPS > 5. Results showed that nivolumab with chemotherapy was superior in mOS (HR 0.71, 95% CI, 0.59–0.86) and in mPFS (HR 0.68, 98% CI, 0.56–0.81) in the PD-L1 CPS > 5 population. Furthermore, mOS (13.8 vs. 11.6 months; HR 0.80, 99.3% CI, 0.68–0.94) in the overall population (not selected for PD-L1) was also superior in the experimental arm. About 60% of the trial enrollment comprised the PD-L1 CPS > 5 group, which may have impacted the positive results in the overall population. Incidence of grade 3–4 adverse events (59% vs. 44%) as well as serious adverse events leading to discontinuation (38% vs. 25%) were higher in the experimental arm compared to the control arm [33]. Also reported at ESMO 2020 was the KEYNOTE-590 phase III trial, which randomized 749 patients to pembrolizumab chemotherapy or chemotherapy alone. Data showed that pembrolizumab with chemotherapy was associated with superior ORR (45.0% vs. 29.3%, *p* < 0.0001), mOS (12.4 vs. 9.8 months; HR 0.73, 95% CI, 0.55–0.76) and mPFS (6.3 vs. 5.8 months; HR 0.65, 95% CI, 0.55–0.76) over chemotherapy alone in the overall, non-PD-L1 selected population [6].

Recent studies have also demonstrated that checkpoint inhibitors improve survival when added to frontline chemotherapy with trastuzumab in HER-2-positive advanced GEC. A recent open-label, single-arm, phase II trial (NCT02954536) enrolled 37 treatment-naïve patients with HER-2-positive GEC to receive pembrolizumab with trastuzumab with chemotherapy. The study demonstrated a six-month PFS of 70% (26/37), meeting its primary endpoint [34]. Additionally, a multicenter, single-arm phase Ib/II PANTHERA trial conducted in Korea recently reported in GI-ASCO 2021 a promising ORR of 76.7%, mPFS of 8.6 months, mOS of 19.3 months, and DOR of 10.8 months [35]. Based on these results, the KEYNOTE-811 trial is currently under way, comparing the addition of pembrolizumab versus placebo to trastuzumab and chemotherapy in untreated, HER-2 positive advanced GEC (NCT03615326).

## 3. Biomarkers Currently Approved for Clinical Use

### 3.1. Mismatch Repair Deficiency/Microsatellite Instability

Germline or somatic mutations in genes encoding MMR proteins such as *MLH1*, *MSH2*, *MSH5*, and *PMS2* as well as epigenetic silencing of *MLH1* lead to an inability of cells to repair mismatched nucleotides during DNA replication which results in MMR deficiency and MSI Figure 1. Consequently, this leads to an increase in neoantigen burden and a heightened response to immune checkpoint inhibition in various tumors. Pembrolizumab was approved on 23 May 2017 regardless of tumor type in advanced dMMR/MSI-H tumors that have progressed on prior treatment [36]. dMMR/MSI-H is determined using several laboratory assays including immunohistochemistry (IHC) to demonstrate the loss of MMR proteins, polymerase chain reaction (PCR) to detect amplification of MSI, and next generation sequencing (NGS) to sequence known MSI genomic loci [37].

Approximately 3% of metastatic or advanced GEC patients harbor MSI [10]. The efficacy of pembrolizumab was assessed in patients with noncolorectal dMMR/MSI-H cancer in the KEYNOTE-158, phase II, basket trial consisting of various solid tumors. The trial enrolled 233 patients across 27 tumor types who progressed on at least one prior systemic therapy. GC was the second most common tumor, making up 10.3% (*N* = 24) of all patients. In this trial, pembrolizumab demonstrated a strikingly high ORR of 34.3% overall in this pretreated population. The results were also impressive in patients with GC with an ORR of 45.8%, mPFS of 11 months; mOS and mDOR were not reached at the time of publication [38]. Furthermore, the NCI-MATCH is a single-arm, multicohort, phase II trial, which enrolled 4902 patients of various relapsed or refractory tumors and assigned to targeted therapies based on molecular testing. The ARM Z1D cohort of this trial comprised 42 patients (*N* = 3 GEC patients) with dMMR/MSI-H non-colorectal cancer, who received nivolumab. The ORR was 36% overall and 66% (PR 2/3) in GEC patients, consistent with the positive results with PD-1 inhibitors in this subpopulation [39]. Additionally, in a phase II trial conducted in Korea, 61 patients who progressed on at least one prior line of therapy received pembrolizumab. Data showed that pembrolizumab was associated with an ORR of 85.7% (6/7) in patients with pretreated MSI GEC tumors [40].

Therefore, dMMR/MSI-H is a robust biomarker of IO response with predictability across various solid tumors and is approved as an agnostic biomarker for pembrolizumab including in GEC. Nonetheless, a significant 40–60% proportion of GEC patients with MMR deficiency/MSI high still do not respond to IO, suggesting that there are additional immunosuppressive mechanisms in effect in the TME.

### 3.2. Tumor Mutational Burden

TMB is defined as the total number of nonsynonymous somatic mutations per coding area of a tumor cell genome and TMB-H is defined as ≥10 nonsynonymous mutations per megabase of the tumor genome. While whole genome sequencing was initially used to measure TMB, methods using NGS are becoming more widely used due to its expediency. On 16 June 2020, pembrolizumab was approved for use in adults and children with TMB-H tumors [41]. This approval was based on the KENOTE-158, phase II, multicohort basket trial. In this study, 1073 patients across various tumor types were enrolled and assessed for TMB and given pembrolizumab. Data published on 10 September 2020 showed that ORR was 29% in TMB-H tumors versus 6% in non-TMB-H tumors. Although the results were impressive, the results reported in this biomarker analysis notably had no GEC patients [38].

Recent studies have reported the association of TMB with response to immune checkpoint inhibition in GEC patients in exploratory analyses. In a phase II trial of pembrolizumab in Asian GEC patients who progressed on at least one prior systemic therapy, TMB was associated with greater ORR. In this trial, TMB-H was defined as >400 nonsynonymous single nucleotide variants. TMB-H patients had an ORR of 88.9% [40]. Additionally, in a phase I/II trial of Chinese GC patients with pre-treated tumors, toripalimab (anti-PD-1 antibody) was associated with higher ORR in TMB-high versus TMB-low patients (33.3 vs. 7.1%, *p* = 0.017). A cutoff of 12 mutations/Mb was defined as TMB-H in this study. This ORR benefit also translated to improvement in OS (14.6 vs. 4.0 mo, *p* = 0.038) [42]. Furthermore, the interim results of the PANTHERA trial reported the exploratory analysis of TMB in patients who received first line pembrolizumab plus chemotherapy with trastuzumab for HER-2-positive GEC. In this analysis, high TMB was associated with a nonsignificant tendency towards increased mPFS (22.0 vs. 8.6 months, *p* = 0.2835) [35]. These results suggest that TMB appears to have a potential as a biomarker of response to anti-PD-1 therapy in GEC.

Of note, controversy exists surrounding TMB’s use as a biomarker, due to its measurement variability across platforms and across tumors and its lack of consensus for a threshold. Therefore, further validation in GEC is needed to determine the feasible cut-off for TMB in GEC.

### 3.3. Programmed Death Ligand-1

Higher levels of PD-L1 expression were associated with poor prognosis in patients with gastric and esophageal cancers in multiple studies [43,44,45,46,47,48]. Such information leads to a logical assumption that higher levels of PD-L1 expression would correlate with better outcomes after PD-1/PD-L1 blockade Figure 1. However, clinical studies show inconsistent results. In a retrospective analysis of ATTRACTION-2, PD-L1 was not a predictor of response to nivolumab in the third-line setting [21]. In the second-line setting of the phase II KEYNOTE-059, patients with PD-L1 CPS ≥ 1 had an ORR of 22.7% (95% CI, 13.8–33.8), while patients with PD-L1 CPS < 1 had an ORR of 8.6% (95% CI, 2.9–19.0) [49]. In the phase III KEYNOTE-061, patients with CPS ≥ 1 had slightly better survival, but patients with CPS ≥ 10 had worse OS than all randomized patients [17]. The phase III ATTRACTION- 3 and KEYNOTE-181 showed marginally higher survival rates in patients with CPS ≥ 1 and CPS ≥ 10, respectively [26,33]. The phase III KEYNOTE-062, previously untreated patients with CPS ≥ 1 had similar survival outcomes to those with CPS ≥ 10, corroborating with the preliminary results from the CheckMate 649 and the KEYNOTE-590 to show that PD-L1 expression is not a very strong predictor of response to ICIs [6,18,33].

The PD-L1 CPS is the most common method to assess PD-L1 expression in gastric cancers [6,18,33,50]. It is a qualitative immunohistochemistry assay, where PD-L1 protein levels are detected in tumor tissues. The CPS is then calculated by dividing the number of PD-L1-positive cells (including tumor cells, immune cells) by the total number of viable cells evaluated, multiplied by 100. Meanwhile, multiple studies in ESCC used the tumor proportion score (TPS), which reflects the percentage of PD-L1 staining relative to all viable tumor cells in the sample, not considering other cells in the TME [26,47,51]. The PD-L1 CPS is currently recommended only for patients with metastatic gastric adenocarcinoma in the United States and is currently not recommended by other major oncology societies worldwide [52,53,54].

For ESCC most of the studies focused on tumor proportion score (TPS) which reflects the proportion of PD-L1 staining only in tumor cells. TPS is also associated with poor prognosis in ESCC patients [47].

## 4. Putative Biomarkers

### 4.1. Epstein-Barr Virus (EBV)

EBV is a human herpesvirus associated with multiple cancer types such as lymphomas, nasopharyngeal carcinomas, and gastric adenocarcinomas. [55,56] Although the carcinogenesis role of EBV is not fully understood, gastric cancers that are positive for this virus show intense intra- and peritumoral immune cell infiltration, also higher levels of PD-L1 expression, making these tumors potentially more responsive to immune checkpoint blockade Figure 1 [9,57]. In a case report, a patient with EBV-positive gastric cancer had a substantial benefit from avelumab [58]. Moreover, a retrospective study also showed better survival in patients with EBV-positive gastric cancers [59]. In 2018, a phase II clinical trial was the first to demonstrate EBV’s clinical significance. Patients with EBV-positive gastric cancer had 100% ORR when treated with pembrolizumab, demonstrating that they might represent a distinct population with a higher potential to respond to ICIs [40].

Indeed, EBV testing has a rising potential, and it could be performed with already established methods such as in situ hybridization or next-generation sequencing [40,60], increasing its implementation feasibility. Still, the current evidence to support this hypothesis is shallow. Hence, EBV testing is currently not recommended for routine oncology care, and there is a need for more extensive prospective trials to consolidate its correlation with clinical outcomes [53,61].

### 4.2. DNA Damage Response (DDR)

Besides MMR, there are four DDR pathways that maintain genomic integrity which include HR, nonhomologous end joining (NHEJ), base excision repair (BER), and nucleotide excision repair (NER) [62]. Based on the effects of MMR deficiency on IO efficacy, it can be reasonably hypothesized that deficiencies in non-MMR DNA repair mechanisms may have a similar effect on IO efficacy. However, the evidence for non-MMR DDR markers’ ability to predict IO efficacy is overall sparse and inconsistent with limited prospective data compared to that for MMR.

The results of a genomic profiling analysis of tumor samples from 17,486 patients with GI cancers including 1750 GC and 2501 EAC patients done using 10 predefined DDR genes (ARID1A, ATM, ATR, BRCA1, BRCA2, CDK12, CHEK1, CHEK2, PALB2, and RAD51) was recently reported. The prevalence of DDR alterations was 17% in the overall sample population, 27% in GC patients, and 19% in EAC patients. Among them, the most frequently altered genes were ARID1A (9.2%), ATM (4.7%), BRCA2 (2.3%), BRCA1 (1.1%), and CHEK2 (1.0%). The prevalence of MSI-H and TMB-H in DDR altered tumors were 19% and 21% respectively. Furthermore, TMB-H (≥ 20 mut/ Mb) was more prevalent in DDR altered cases even in MSS tumors in the overall sample population. Among cases with DDR altered and TMB-H tumors, 87% were also MSI-H. Taken together, these results suggest DDR alterations are associated with increased neoantigen burden and that mutual exclusivity exist among MSI-H, TMB-H, and non-MMR DDR deficiencies [63].

A pan-cancer cohort study was conducted in patients with advanced tumors treated with immunotherapy. NER and HR deficiencies were found in 3.4% and 13.9% of patients respectively and about 20% of GEC patients harbored deficiencies in HR, NER, or both. Mutations in NER (adjusted hazard ratio (aHR) 1.59, 95% CI 1.10–2.30) and HR (aHR 1.39, 95% CI 1.15–1.70) associated with prolonged survival in patients who received checkpoint inhibition but not in patients who did not receive checkpoint inhibition after adjustment for TMB, tumor type, and checkpoint inhibitor type. OS benefit was additive, in that mutations in both NER and HR pathways resulted in an even greater OS benefit above mutations in one of the pathways. ORR benefit was also statistically greater for mutant NER or HR in the GEC patients [64].

*POLE* and *POLD1* are genes that encode DNA polymerase epsilon and delta respectively. While these genes are not involved in DDR responses, they are crucial in DNA replication proofreading and fidelity and mutations in either gene is associated with increased TMB. A pan-cancer analysis of *POLE/POLD1* mutations in 47,721 patients showed that 185/2586 esophagogastric cancer patients harbor such mutations. Analysis in the overall population in patients who received ICI showed OS was significantly longer in patients with either *POLE* or *POLD1* mutations (median OS, 34 months) compared to wild type patients (18 months). Among those with *POLE/POLD1* mutations, 74% were MSS. *POLE/POLD1* mutations remained an independent risk factor for greater response to immune checkpoint inhibitor after adjusting for MSI status and cancer type. These results warrant the evaluation of *POLE/POLD1* mutations in future prospective trials in GEC [65].

### 4.3. Gene Expression Profiles

GEP is a group of genes in a cell with a characteristic pattern of gene expression usually indicative of immune responses in the context of checkpoint inhibition Figure 1, Table 2. In melanoma patients, a GEP indicative of interferon gamma signaling was shown to be predictive of improved ORR and PFS to PD-1/PD-L1 blockade [66]. Additionally, in non-small-cell lung cancer and head and neck squamous cell cancer GEPs have found to be associated with response to PD-1/PD-L1 blockade [67]. Several trials have also evaluated the association between GEP and immunotherapy in GEC.

In the KEYNOTE-012 trial, an 18-gene interferon gamma signature panel that was correlated with IO response in melanoma was studied. Although the association between the GEP and ORR was statistically nonsignificant, there was a tendency towards improvem1ent in patients with high GEP scores [13]. In the subsequent KEYNOTE-59 study, the same GEP was found to be associated with ORR to pembrolizumab [15]. Additionally, a distinct four-gene inflammatory signature panel was also associated with response to nivolumab in CheckMate-032 [20].

An enriched B cell gene signature of TME associated with higher clinical benefit rate (CBR), PFS and OS in ESCC patients receiving PD-1/L-1 blockade [68]. Also, a 24-gene RNA signature derived from immune related gene expression profiling constructed using machine learning was tested in a GI cancer population. The study consisted of 96 patients in two separate cohorts comprising 40% GC and 26% EC patients. Immune checkpoint inhibitor treated patients were stratified into either high or low scoring groups based on the Youden index of a receiver operating curve (ROC) curve. By comparison of the area under the curve of the ROC curve (AUC), the RNA signature outperformed PD-L1, TMB, and MSI status [69]. In contrast, in a phase Ib/II trial of second line durvalumab with/without tremelimumab in GEC patients, a tumor RNA-based IFNγ signature was not associated with improved clinical response [24]. This suggests that the predictive potential of GEP signatures may vary based on type of checkpoint inhibitor agent and/or histological subtype of GEC.

Overall, further investigation in large phase III trials is needed for the validation of GEP for use with immunotherapy in GEC.

## 5. Conclusions

Besides the tumor agnostic markers MSI and TMB, the only validated immunotherapy predictive biomarker in GEC is PD-L1 CPS, which is useful in limited settings and lacks robustness. Thus, novel biomarkers need to be developed for better patient selection. Important questions in the future will be to determine the role of immunotherapy biomarkers for combination therapies, in particular, whether a given biomarker will retain its predictive ability when immunotherapy is used with therapies with distinct nonoverlapping mechanisms of action. Another important question will be whether immunotherapy biomarkers will be able to predict synergy between the agents that will be used in combination. Because of increased toxicities, development of robust biomarkers and improved patient selection will be keys for combination immunotherapies. Furthermore, clinical trials are under way to assess immunotherapy-based neoadjuvant or adjuvant therapy in locally advanced GEC [71,72,73,74]. Whether the predictability of biomarkers already validated in advanced stage tumors will translate to earlier treatment settings will be another important question to answer in ongoing and future trials.

## Figures and Tables

**Figure 1 cancers-13-01715-f001:**
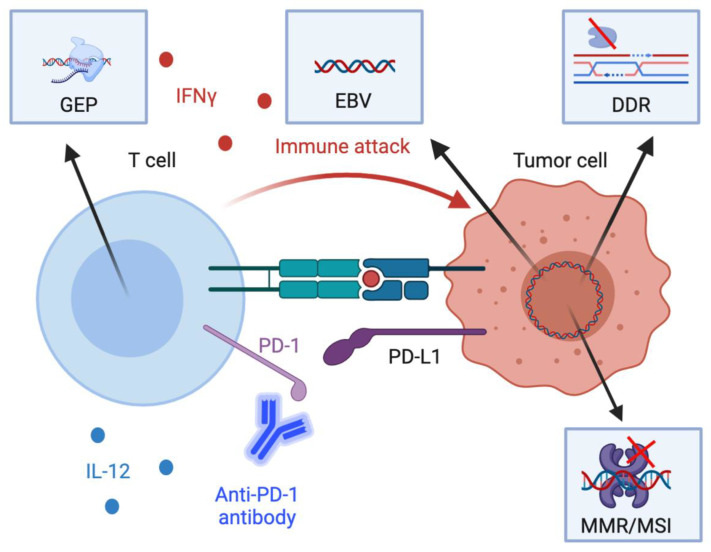
Schematic diagram of the mechanistic basis of key biomarkers. Several available putative and established biomarkers reflect both tumor-intrinsic and –extrinsic processes that influence anti-tumor immunity. First, engagement of programmed death-1 (PD-1) on CD8+ T cells by PD-L1 on tumor cells and other cells in the tumor microenvironment leads to inhibition of anti-tumor cytotoxic T cell responses. PD-L1 expression in tumor cells, lymphocytes, and macrophages in tumor specimens are associated with heightened immunotherapy response. Second, interferon gamme (IFNγ) and interleukin-12 (IL-12) in the tumor microenvironment activate T helper type 1 responses which are associated with heightened cell-mediated cytotoxicity. Gene expression profiles associated with IFNγ are associated with immunotherapy response. Third, defects in the DNA damage response (DDR) as well as the mismatch repair response (MMR) are associated with a greater tumor mutational burden. Additionally, Epstein-Barr Virus (EBV) positivity reflects the expression of viral antigens on tumor cells, which are highly immunogenic. Both the increased tumor mutational burden from DDR defects and MMR deficiency and EBV positivity thus are associated with greater immunotherapy response. Other acronyms: GEP, gene expression profiles; MSI, microsatellite instability. Created with BioRender.com (Accessed: 15 March 2021).

**Table 1 cancers-13-01715-t001:** PD-L1 in selected clinical trials.

Trial ID	Tumor	Agent	Line	Phase	PD-L1	N	HR OS (95% CI)	HR PFS (95% CI)
KEYNOTE-061	G/GEJ	Pembrolizumab	2nd	III	All	592	0.94 (0.79–1.12)	1.49 (1.25–1.77)
					CPS ≥ 1	395	0.82 (0.66–1.03)	1.27 (1.03–1.57)
					CPS ≥ 10	197	2.05 (1.5–2.79)	NA
ATTRACTION-3	ESCC	Nivolumab	2nd	III	All	419	0.77 (0.62−0.96)	1.08 (0.87–1.34)
					TPS ≥ 1%	203	0.69 (0.51−0.94)	NA
KEYNOTE-181	EAC/ESCC	Pembrolizumab	2nd	III	All	628	0.89 (0.75–1.05)	1.11 (0.94–1.31)
					CPS ≥ 10	222	0.69 (0.52–0.93)	0.73 (0.54–0.97)
ESCORT	ESCC	Camrelizumab	2nd	III	All	448	0.71 (0.57–0.87)	0.69 (0.56–0.86)
					CPS ≥ 1%	191	0.58 (0.42–0.81)	NA
CheckMate 649	G/GEJ/EAC	Nivolumab plus CT	1st	III	All	1581	0.8 (0.68–0.94 ^a^)	NA
					CPS ≥ 1	1296	0.77 (0.64–0.9 ^a^)	NA
					CPS ≥ 5	955	0.71 (0.59–0.86 ^b^)	0.68 (0.56–0.81 ^d^)
ATTRACTION-4	G/GEJ	Nivolumab + CT	1st	III	All	724	0.9 (0.75–1.08)	0.68 (0.51–0.90 ^c^)
KEYNOTE-062	G/GEJ	Pembrolizumab	1st	III	CPS ≥ 1	506	0.91 (0.74–1.10)	1.66. (1.37–2.01)
					CPS ≥ 10	182	0.69 (0.49–0.97)	1.1 (0.79–1.51)
		Pembrolizumab + CT			CPS ≥ 1	507	0.85 (0.70–1.03)	0.84 (0.70–1.02)
					CPS ≥ 10	189	0.85 (0.62–1.17)	0.73(0.53–1)
KEYNOTE-590	ESCC/EGJ	Pembrolizumab + CT	1st	III	All	749	0.73 (0.62–0.86)	0.65 (0.55–0.76)
					CPS ≥ 10	NA	0.62 (0.49–0.78)	0.51 (0.41–0.65)
							**ORR (95%CI)**	
KEYNOTE-180	EAC/ESCC	Pembrolizumab	4th+	II	All	121	9.9% (5.2–16.7)	
					CPS ≥ 10	58	13.8% (6.1–25.4)	
					CPS < 10	63	6.3% (1.8–15.5)	
KEYNOTE-059	G/GEJ	Pembrolizumab	3rd+	II	All	259	11.6% (8.0–16.1)	
					CPS ≥ 1	148	15.5% (10.1–22.4)	
					CPS < 1	109	6.4% (2.6–12.8)	

ESCC = esophageal squamous cell carcinoma; G = gastric; GEJ = gastroesophageal junction; EAC = esophageal adenocarcinoma. ^a^ 99.3% CI, ^b^ 98.4% CI, ^c^ 98,51% CI, ^d^ 98% CI.

**Table 2 cancers-13-01715-t002:** Selected trials with immune biomarkers other than PD-L1.

Trial	Number of Previous Lines of Treatment	Treatment	Biomarkers Tested
KEYNOTE-028 [14]	Any	Pembrolizumab	GEP
KEYNOTE-012 [13]	Any	Pembrolizumab	GEP
RiME (NCT03995017)	1–2	Rucaparib + ramucirumab +/− nivolumab	Homologous recombination deficiency
NCT02915432 [42]	≥1 (cohort 1) or 0 (cohort 2)	Toripalimab +/− chemotherapy	TMB
NCT02589496 [40]	≥1	Pembrolizumab	MSI, TMB, EBV
PANTHERA [35]	≥1	Pembrolizumab + chemotherapy	TMB
NCT02340975 [24]	≥1	Durvalumab +/− tremelimumab	MSI, GEP
KEYNOTE-061 [70]	≥1	Pembrolizumab	MSI
CheckMate-032 [20]	≥1	Nivolumab	MSI
KEYNOTE-059 [49]	≥2	Pembrolizumab	MSI, GEP

GEP, gene expression profile; TMB, tumor mutational burden; MSI, microsatellite instability; EBV, Epstein–Barr virus.

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
