# Peer review of "Immunotherapy Predictive Molecular Markers in Advanced Gastroesophageal Cancer: MSI and Beyond"

_cancers, 2021, doi:10.3390/cancers13071715_

Round 1
Reviewer 1 Report
The manuscript by Park et al. reviewed the immunotherapy predictive molecular markers in advanced gastroesophageal cancer. The review is thorough and quite exhaustive. However, addressing a few minor issues would further enhance the impact of the review.
- Instead of using continuously numbered headings, the authors should organize the review differently. After the introduction, the authors should use a heading introducing the biomarkers. This section will have two subheadings: biomarkers currently approved for clinical use and putative biomarkers (these are already present in the last paragraph of the introduction, but putting these under a separate heading will make it easier for the readers).
- The authors should utilize the figure more. It would be more helpful if the authors referred to the figure during the discussion. When discussing different biomarkers, the authors should specifically mention Figure 1. For example, the author should refer to Fig 1 section A (anti-PD-1 antibody) when discussing the use of checkpoint inhibitors.
- It will also help the reader if the authors very briefly summarize how checkpoint inhibitor (and anti-CTLA4) therapy works in cancer.
Overall, the authors should be commended for writing a comprehensive review on a topic that is becoming increasingly important.
Author Response
- Section headings have been edited as the reviewer had recommended.
- As the reviewer has recommended, Figure 1 has been commented on in the body of the manuscript.
- A section explaining the mechanism of checkpoint inhibitors has been added: line 95-99.
We appreciate the reviewer's thoughtful comments.
Reviewer 2 Report
I wish to congratulate to the Authors for the comprehensiveness and clarity of the text and for the mastering of syntax in writing (unfortunately, an infrequent occurrence in scientific articles). Nothing to add. Few minor improvements of text are listed as follows.
1) Legend to Figure 1 - it should include the explanation of two acronyms: “GEP, gene expression profile; MSI: microsatellite instability”;
- end of Line 87: “activate”, not “activates”;
- line 88: “cell-mediated”, not “cell mediated”.
2) Line 97 - “PD-L1-positive”; the attribute of an adjective should be always connected to the latter by a dash (like in the case of “cell-mediated”); the point applies to several other
similar cases throughout the text (e.g. “progression-free survival”, line 128).
3) Line 103 - remove “this” from the beginning of line.
4) Lines 208/210/216 - I would suggest to homogenize the writing to “HER2-positive”.
5) Line 319 - it should read “intra- and peri-tumoral”.
6) Line 334 - “DNA damage response (DDR)” is already defined (lines 77-78): change to “DDR”.
Author Response
- Explanation of the acronyms GEP and MSI have been added to the figure legend. Activates has been changed to activate. Cell mediated has been changed to cell-mediated.
- The specified grammatical errors have been addressed.
- "This" has been removed.
- HER2 positive has been changed to HER-2-positive.
- Peritumoral has been changed to peri-tumoral.
- DNA damage response has been changed to DDR.